# Identifying engagement strategies for Hispanic youth with anxiety: A youth-centered, Design-Thinking approach

Thea Runyan[1], Kristen Hassmiller Lich[1], Karl Umble[1], Karen Li[2], Kiersten Daniel Baca[3], Leah Frerichs[1]*

**1** Department of Health Policy and Management, UNC Gillings School of Public Health, University of North Carolina, Chapel Hill, North Carolina, United States of America, **2** Director of School Health, Sequoia Healthcare District, Redwood City, California, United States of America, **3** Independent Consultant

* leahf@unc.edu, leahf@email.unc.edu

**Data Availability Statement:** All relevant data are within the paper and its Supporting information files.

## Abstract

Anxiety is the most common and widespread mental health disorder impacting youth between the ages of 10–19. Youth of color including Hispanic youth are disproportionately impacted. Fewer than 20% of youth of color who need mental health services are receiving them. However, we know relatively little about how to best engage Hispanic youth to increase their use of mental health services. The aims of this study were to better understand the personal, environmental, and behavioral factors that impact Hispanic adolescents help-seeking behaviors and to identify the important criteria needed to develop an appealing intervention that would increase engagement with mental health services. This study used a Design Thinking process—a participatory research approach that included qualitative and engaged methods. In-depth interviews (n = 8) and a Design Thinking workshop (n = 11 participants) were conducted with Hispanic youth with anxiety residing in the San Francisco Bay Area. In-depth interviews were coded using Social Cognitive Theory to identify key themes that impact an adolescent's decision to seek help. The 90-minute workshop included ideation and Design Thinking activities including personas, mind-mapping, and analytical problem-solving to identify the most important tools and strategies that could be used to manage anxiety. The study identified several themes that directly impact program design, including barriers to seeking help for anxiety, coping strategies, sources of support, and specific program ideas. The findings revealed that Hispanic youth want a culturally relevant technology-based program that provides easily accessible educational information and coping strategies delivered in an engaging format that also facilitates mental health support with a trusted adult. The results reinforce the need to develop culturally inclusive and innovative programs designed specifically for priority populations to increase youth engagement with mental health services.

**Funding:** The author(s) received no specific funding for this work.

**Competing interests:** The authors have declared that no competing interests exist.

## Introduction

Anxiety is the most common and widespread mental health disorder impacting children and adolescents [1]. The Diagnostic and Statistical Manual of Mental Disorders, 5th ed. (DSM-5) describes anxiety as excessive and uncontrollable worry about social events or daily activities [2]. While it is a common emotion that is usually experienced as a temporary feeling when anticipating certain events such as an important test or preparing for a job interview, [3] for many people anxiety is not just a passing worry or fear. People with anxiety experience ongoing symptoms (e.g. restlessness or irritability, difficulty sleeping, fatigue, muscle tension, shortness of breath, increased heart rate, trouble concentrating) that are difficult to manage even when the event is over [2, 4]. The physical, social and emotional impact on individuals with anxiety can be significant [5]. Adolescents suffering from anxiety are more likely to have challenges with school and interpersonal relationships, which may lead to poor social and coping skills, low self-esteem, decreased academic success and increased rates of depression [5]. Since anxiety is also considered a chronic condition, there are often longer-term consequences lasting well into adulthood.

Anxiety impacts approximately one in three (31.4%) US adolescents between the ages of 13 and 18 [3]. Hispanic youth, who represent 25% of the children in the US [6], are disproportionately impacted with higher rates of mental health disorders than their black and non-Hispanic white peers [7–11]. Even though Hispanic youth are at greater risk, they are far less likely to receive mental health services [9, 12–14]. Fewer than 20% of Hispanic youth are receiving the mental health support they need, and if they do get treatment, they are more likely to drop out earlier than their non-Hispanic peers [14].

Many structural barriers prevent youth from accessing mental health support, for example, lack of primary care providers trained to diagnose mental health issues, a shortage of mental health providers who have experience working with youth of color, transportation and geographic isolation, inadequate or no insurance coverage, and the inability to pay for services [15, 16]. These are exacerbated by psychosocial barriers, such as cultural attitudes and beliefs that stigmatize mental health issues, general mistrust of authority and the health care system, negative familial beliefs about mental health disorders, fear of being judged, language barriers, and negative attitudes towards therapists and healthcare providers [9, 13, 15, 17, 18]. Race-based discrimination from medical providers and lack of access to quality, evidence-based health care services are also reported as barriers to treatment [9, 12, 16]. If left untreated, anxiety can lead to more serious mental health problems such as depression and substance misuse in adulthood [3, 5, 14, 19].

To increase adolescent utilization and engagement with mental health services, it is important to consider alternative, and potentially more acceptable, methods of treatment [20]. Technology-based interventions such as online medical visits, health programs delivered via computer, internet, video games or smartphones are examples of innovative strategies that reduce many of the barriers that prevent adolescents from seeking and engaging in treatment for anxiety. While most technology-based interventions for mental health have been developed and researched for adults, the few studies on anxiety treatment in adolescents show effectiveness, though youth engagement and completion rates remain very low [21–24]. In addition, the majority of participants in these studies have been white, limiting our understanding about the acceptability of technology-based interventions among Hispanic youth in the US.

The goal of this research was to understand the factors influencing youth engagement with mental health services, particularly the behaviors, attitudes and structural barriers that impact Hispanic youth's decision to seek help for anxiety and stay engaged in treatment. To achieve this, the study used Design Thinking, a solutions-oriented methodology emphasizing

empathy, co-creation, and iterative problem-solving. Design Thinking was selected as the research methodology because of its strong emphasis on understanding human behavior and promoting meaningful engagement by focusing on the needs and preferences of potential users of the programs or product [25]. It is a structured and flexible research process that aligns well with youth participatory research and is particularly effective for developing innovative solutions. The aim was not to create a complete intervention but to identify key engagement strategies and design criteria to inform future program development. This paper presents the findings from key informant interviews and a Design Thinking workshop with Hispanic youth experiencing anxiety.

## Materials and methods

### Study design

This study used a Design Thinking approach, incorporating qualitative and participatory research methods to explore engagement with mental health services among Hispanic youth with anxiety. Design Thinking is a widely used participatory human-centered methodology that has been applied in healthcare, education, and business to solve complex problems. The framework involves five iterative stages: Empathize, Define, Ideate, Prototype, and Test [26]. While these stages are often presented linearly, the expectation is that there is continuous iteration throughout the research process. For this research we focused on the first three stages: *empathize*, *define* and *ideate* to gather data that could inform the design of interventions to improve youth engagement with mental health support. Table 1 outlines the qualitative and participatory methods that were used in the first 3 Design Thinking stages. The study was approved by the Institutional Review Board at the University of North Carolina at Chapel Hill. Oral and written consent were obtained from study participants and their parents.

**Stage 1: Empathize.** Key informant interviews with Hispanic adolescent students were conducted to obtain a deep understanding of the person the program is being designed for (adolescents), the challenges they face in doing the desired behavior (seeking treatment for anxiety), and what they need (resources, support) to do the behavior in the future (receive treatment). At the end of each interview, participants were asked to suggest specific program features they believed would motivate teens to adopt coping strategies and actively engage with an intervention addressing their anxiety.

**Stage 2: Define.** The objective of this stage was to define the fundamental problem to be solved. A literature review reinforced the gap between adolescents' need for mental health support and their lack of utilization and engagement with services. The data collected from the literature review and the key informant interviews were synthesized into a *problem statement* and a *persona*. The problem statement clearly states the problem that needs to be solved. The persona is a fictional character based on the research. It is used to define and represent the problem in a way that is relatable to the people trying to solve it. It is usually

**Table 1. Initial three stages of Design Thinking and qualitative methods used for the study.**

| Stage in Design Thinking Process | Stage 1 Empathize | Stage 2 Define | Stage 3 Ideate |
|---|---|---|---|
| Purpose | Seek to Understand | Define the Problem | Share Ideas & Prioritize |
| Qualitative methods | **Key Informant Interviews** to uncover barriers and motivating program features | **Thematic Analysis of Interviews** to define the problem | **Design Thinking Workshop** to generate and prioritize ideas |
| Associated Design Thinking activity | Key Informant Interviews | Problem Statement & Persona | Mind Mapping |

presented with a photo and a short description of the persona's behavior patterns, goals, skills, attitudes and environment as they relate to the problem statement. In this study, the persona is an archetype that represents the youth who will ultimately use the program that is being designed [27, 28]. The *problem statement* and the *persona* were used during the Design Thinking workshop to guide the participants in their exploration of strategies to promote coping and engagement with mental health support.

**Stage 3: Ideate.** The objective of this stage was to engage participants in generating and prioritizing ideas for strategies that could improve youth engagement in mental health support. To do this, we conducted a Design Thinking workshop with a group of adolescents who worked collaboratively through a series of interactive activities. Using the problem statement and persona developed in Stage 2, participants engaged in creative thinking and analytical problem-solving tasks. For instance, participants brainstormed practical coping strategies to help the persona manage anxiety and discussed barriers that might prevent engagement with these strategies or mental health support. Participants then voted on the barriers they considered most challenging and brainstormed potential solutions to help the persona overcome these obstacles and access support. As a final activity, they created individual mind-maps around key themes identified in the key informant interviews.

## Research setting and participant selection

Key informant interview participants were selected through purposeful sampling of Hispanic youth who self-reported that they had experience with anxiety. Recruitment took place through schools and community organizations in the San Francisco Bay Area. Potential participants were identified with a referral from their school counselor, teacher, or therapist. To determine eligibility, interested participants received a flyer about the study and were asked to contact the researcher via email if they wanted to participate. Potential participants received an official email invitation with a brief description of the study using a standardized script. They were also sent a copy of the youth assent form and parental consent form that explained inclusion and exclusion criteria. To be included, potential participants had to be between the ages of 13 and 17, identify as Hispanic, and report feelings of anxiety. Exclusion criteria for student participation included feelings of self-harm, suicide ideation, or depression in the past 12 months. Importantly, no student reported these conditions. This exclusion criterion reinforces that the study population was not severely ill, and the focus of our research was specifically on identifying coping strategies and sources of support for adolescents managing anxiety rather than those experiencing more severe mental health challenges. If the student met the inclusion criteria, they were asked to complete the consent forms and schedule a video interview with the researcher for a date and time that was convenient for them.

Participants in the Design Thinking workshop were identified through a community outreach coordinator at a local high school. The outreach coordinator also helped obtain parental consent and secured the workshop time and space which was held after school in the school conference room. The workshop participants were exempt from IRB approval because they were volunteers and were not asked to provide any personal information.

## Data collection and analysis

**Key informant interviews.**   The questions developed for the key informant interviews were based on the Social Cognitive Theory (SCT) [29, 30]. The questions focused on the

personal and environmental factors that influence an adolescent's help-seeking behaviors. In this model of Bandura's SCT [30] which was adapted for youth mental health (Fig 1), the personal factors included the youth's personal experience with mental health issues and treatment, gender, ethnicity, socio-economic status, and self-efficacy—the belief that they have control over their choices. The environmental factors included access to services, the impact of the home and school environment, the influence of family and friends, and the societal stigma associated with mental health. The behavioral factors included the knowledge and skills they already have or need to acquire to manage their anxiety, their expected outcomes from participating in treatment and the positive reinforcements that help motivate behavior change.

The interview guide was derived from the theoretical framework to collect detailed information about the following research questions:

- When we talk about anxiety, what does that mean to adolescents? Are they aware of anxiety in themselves and their friends? What does anxiety look like or feel like to them?

- When, Where and Why do adolescents seek treatment? At home, school, therapist's office?

- How do adolescents want to receive treatment? On a computer, phone, or in an office?

- What do adolescents do now to cope with feelings of anxiety? Who do they go to for support?

- Who do adolescents want to guide their treatment? A therapist, coach, peer, self, parent?

The key informant interviews took place over a video platform called Zoom. Field notes were taken during each interview. The interviews were recorded and the audio files were transcribed using the Zoom transcription service. The data were thematically analyzed with deductive and inductive coding [31]. A deductive approach was used to apply broad codes to the data based on the SCT framework and interview questions (i.e., causes of anxiety, resources, sources of support). The transcripts were uploaded to a qualitative and coding analysis software program called Delve. Transcripts were reviewed again with the application of deductive and inductive codes to identify participant quotes and descriptive statements with relevant

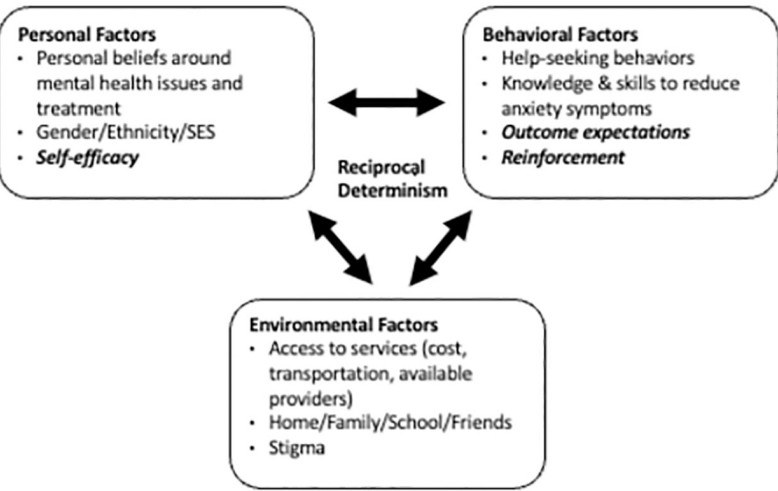

**Fig 1. Social cognitive theory [29, 30] adapted for the study.**

qualities and characteristics. After coding was completed, patterns were identified and used to condense and summarize the codes into themes and sub-themes. A codebook was developed with the key themes, sub-themes and their definitions. To ensure reliability of coding, a second researcher who was not involved in the study independently reviewed and coded 10% of the transcripts using the codebook. The two coders discussed differences in coding to reach a consensus.

After the coding was validated, the frequency of themes (i.e., how many participants mentioned each theme) was calculated. In the majority of interviews, the significance of the coded themes and sub-themes was very clear with half of the responses supporting them, however, sub-themes with fewer responses were considered significant if they were unique and introduced relevant and interesting discussion points. The frequencies of themes were also used to develop the persona by highlighting common behavior patterns, goals, skills, attitudes and environment among the youth.

**Design Thinking workshop.** A Design Thinking workshop was conducted with a group of Hispanic high school students after school in the school's conference room. All students who participated in the key informant interviews were invited to participate in the Design Thinking workshop. The goal of the workshop was to engage a group of adolescents in structured activities designed to generate new ideas and creative solutions that would inform the program's development and increase engagement in anxiety treatment.

At the beginning of the workshop, participants were asked to review the persona and problem statement. The fictional persona, called Sofia, was the archetype constructed from the data collected from the key informant interviews to represent an adolescent's experience with anxiety, the strategies used to deal with their anxiety, and their help-seeking process. The fictional persona was presented to participants with a short description of her behavior patterns, goals, skills, attitudes, and environment as they related to the problem statement (Fig 2). The problem statement, "teens who need support for anxiety are not engaging in treatment", clearly stated the problem that needed to be solved. The persona and problem statement were used to guide the participants through a series of brainstorming activities designed to generate ideas and considerations that could influence youth engagement with mental health support and potentially improve their mental health journey. Keeping Sofia, the persona, in mind, participants were asked to vote on the generated ideas to prioritize the factors most likely influence her help-seeking behaviors for anxiety support. The data from the brainstorming activities were recorded on post-it notes that were collected at the end of the workshop and entered into a spreadsheet. In the spreadsheet, the data was reviewed and categorized into broad themes

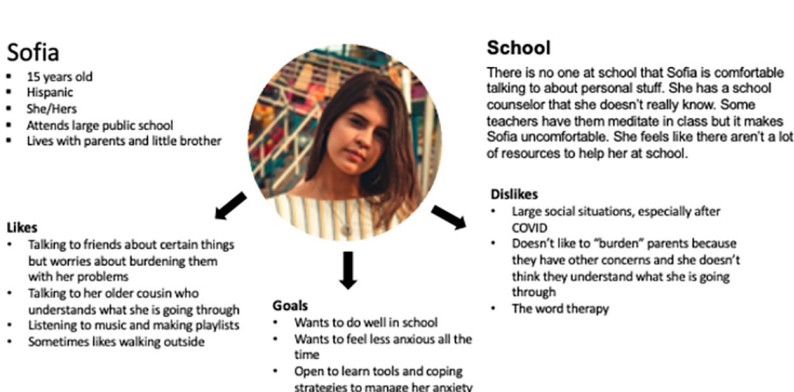

**Fig 2. Persona presented to students in the Design Thinking workshop.**

using an inductive approach based on the student responses. The frequency of student responses for each theme was used to determine which themes were most common (themes with equal to or greater than 5 responses), however, all responses were considered.

For the final workshop activity, students were asked to create a mind-map, which is a visual tool to help inspire new ideas and insights [25]. Each student received a blank piece of paper and was asked to write the word "Anxiety" in the middle of the paper with four lines coming out of the center. Each line was associated with a key construct of the SCT and a central theme from the key informant interviews that impact an adolescent's decision to seek support for their anxiety. The four themes were: Technology use (a Behavior construct), School (an Environment construct), People Who Support Me (a Personal construct), and Coping Strategies (a second Behavior construct). The students were asked to brainstorm around these four themes considering Sofia, the persona, as the person who would be seeking mental health support. Specifically, they were asked to write down what comes to mind for each theme without consulting with their peers. After the workshop, the mind maps were collected anonymously and combined into one comprehensive meta-map using an infographics software called Venngage. The meta-map provides a visual representation of the student responses showing the frequency and variety around each of the four themes. For example, a larger circle means there were many of the same responses and a diamond shape indicated a negative association with the theme.

## Results

### Key informant interviews

A total of eight adolescents between the ages of 14 and 17 participated in key informant interviews. All eight participants identified as female and of Hispanic origin. Seven of the participants attended the same charter high school with a population of 400 students, of which 80% identified as Hispanic. One participant was enrolled at a large public high school that had a population of 1800 students of which 60% of students identified as Hispanic. The average length of the interviews was 41 minutes. Each student received a $25 gift card for participation.

**Interview themes.**   The results of the interviews were organized around 8 themes and 22 corresponding subthemes that emerged from the data analysis (S1 Table). A summary of all themes, sub-themes with descriptions and key findings are presented in Table 2. Additional detailed findings are reported for the following four themes: barriers to seeking help, coping strategies, sources of support, and program ideas, which have a direct impact on program design. Participant quotes are included to provide additional context and pseudonyms were used to protect each participant's identity.

**Barriers to seeking help.**   The participants were asked why they did not want to get help for their anxiety. Almost all the participants said that they did not trust many people to share their feelings with and were afraid of being judged, even by family and friends. In addition, despite a few positive experiences with therapy, most students had a negative perception of therapy and therapists that prevented them from seeking help.

*Lack of trust.* Seven students said that they did not want to get help because they did not want to talk about their feelings with someone they did not know or trust. For example, Alejandra (all names are pseudonyms) said she would have to "really trust that adult to be able to go to them for being just stressed" and she did not like the idea of other people "knowing your problems." Lucia said she did not trust people because she had a negative experience in the past. She said she felt "betrayed I guess like a lot of times" so she decided she did not want to share her feelings with other people. Olivia did not want to talk to someone for support

**Table 2. Summary of themes and key findings from student interviews.**

| Theme | Sub-theme(s) | Description | Key Findings |
|---|---|---|---|
| **Causes of Anxiety** | School, Social Situations, Social Apps | What Hispanic youth believe to be the most common causes of anxiety. | School issues such as academics and interpersonal relationships, social situations, social media. |
| **Barriers to Seeking Help** | Lack of Trust, Fear of Judgment, Attitude Towards Therapists | The psycho-social and structural barriers that impact a student's decision to seek help. | Lack of trust of people they do not know and fear of being judged by others, negative attitude toward therapists. |
| **Coping Strategies** | Physical Activity, Journaling, Social Media, Music | Coping strategies that adolescents use to help manage their feelings of anxiety. | Physical activity, journaling, social media as a distraction, listening to music and making playlists. |
| **School Resources** | Wellness Rooms, School Counselor, Teacher/Mentor, School Therapist | Resources available at school to help students when they are feeling anxious. | Wellness rooms for breaks during the school day and teacher/mentor. School counselors and therapists were mentioned even though they were not accessible to all students. |
| **Sources of Support** | Trusted Adult, Friends, Family Members, Self, Therapy | The people that adolescents go to for help and support when feeling anxious. | Friends, trusted adult, family members (e.g., parent, cousin, sibling), self. Therapy was the only external resource mentioned by teens a source of support. |
| **Program/ Product Ideas** | Education & Coping Skills, Interactive Games & Activities, Access to a Trusted Adult | Ideas from adolescents of what they and their peers want in a program designed to help manage their anxiety. | Education and coping skills, interactive games and activities, and access to a trusted adult. |
| **Things to Consider** | Not Applicable | Extra advice provided on how to engage adolescents in a new program. | Avoid certain words, provide direct and accurate information, incorporate a social media influencer. |

because she did not trust the information would be kept confidential. She was worried about people spreading "rumors" and then "everyone knows what is going on."

*Fear of judgment*. Several participants said that they were afraid people would judge them negatively if they shared their feelings. Maya said that even though she was close to her brother "we are not on that level of close." She would not share her feelings with him because she was worried "he might laugh at me or something like that." Several students were concerned about being judged by their parents. Maya referred to the cultural barriers that prevented her from talking to her parents. She said that she did not talk to her parents because "they are Mexican, they don't believe in anxiety or depression." She also did not feel like she could explain it to them because they would either blame her for "being lazy" or blame it on the time she spent on her phone. Lucia was worried about being judged by her friends. She said that no matter how close you are, "some part of them is going to judge you." Olivia was also worried about how her friends might react so she did not like to share her feelings because she did not want them to see her as "vulnerable."

*Attitude towards therapists*. Six participants said that they had a negative opinion of therapy and therapists, which prevented them from going to therapy. Some of the negative perceptions were based on personal experience. For example, Maya said that she tried therapy but did not want to go back because "it was just a bunch of dumb questions and I feel like it wasn't benefiting me." Victoria also tried talking to a therapist but could not develop a good relationship with them due to lack of trust. She said she did not go back because "it is just like someone you don't know so you don't want to talk about anything to them." Gabriela had not seen a therapist but developed a negative attitude toward therapy based on what she heard from her friends and family members. Specifically, she said she did not want to go to a therapist because her friend said that therapists were not trying to help them, they were just trying to make their patients "codependent of them" to keep them coming back because "it is the therapist's source of income."

**Coping strategies.** The participants shared a variety of coping strategies that they liked to use to manage their anxiety, however, there were strong themes around physical activity, journaling, social media and music, even if students had a slightly different approach.

*Physical activity.* Participating in some type of physical activity was mentioned as a positive coping strategy by five students. The type of physical activity ranged from actively working out at the gym to taking walks around the school campus or walking in their neighborhood after school. Maya felt the physical and emotional benefits of working out. She said she liked it because "it just honestly makes you feel better." Amelia was one of several participants who said she liked to take walks during the school day when she needed a break at school. She said that her school allowed students to ask for a special pass from their teacher which allowed them to "walk around for a little bit if we are feeling anxious."

*Journaling.* All but one participant said they liked to keep a journal for keeping track of their feelings, however, they all liked to use different self-monitoring methods. Alejandra kept a paper notebook where she would write for a while and then go back to read it later. She said that this helped her process her feelings and "figure out what can I do to have this not be my problem." Several participants said they liked to use the notes app in their phones and lock it for privacy. For example, Lucia shared that she wrote in her notes app when it got "really, really bad" and made sure to "lock it each time" so no one would be able to read it. Gabriela used her notes app when she felt stressed or "wanted to cry." She would "write everything down and type it out" which made her feel better, then she would "just lock the note." Like Alejandra, Gabriela also liked to look at previous journal entries because she thought it was helpful to reflect and "see what I was going through, because I know how to deal with these things now."

*Social media.* Six participants said that they looked at social media, particularly TikTok, which is a short form video sharing app, as a coping strategy to distract themselves from their feelings. Victoria shared that she would go on her phone and "watch TikTok or something" and explained she would do this to avoid "putting attention" to her anxiety. Alejandra said she spent "a lot of time on TikTok" as a distraction and so did Amelia who specifically said she watched TikTok to "distract myself." Maya also said that she watched TikTok videos because the distraction made her "feel less anxious and just more calm."

*Music.* Listening to music was another coping strategy mentioned by several students. Olivia said that she liked to "lay down or listen to music" when she felt anxious. Maya said she listened to music "a lot" because it made her "feel so calm." She said that listening to music helped "alleviate" some of her problems and would help her try "not to worry about stuff."

**Sources of support.** When the participants were asked who they were most likely to go to when they needed support, most of them said they would go to their friends despite concerns about how they might respond. They also said they would go to a trusted adult if one was accessible, or a family member depending on their relationship with that family member. A few participants said that they preferred to keep their feelings to themselves and work it out on their own.

*Friends.* All eight participants said that their friends were a source of support and would often reach out to them first before going to a trusted adult or family member. Amelia said that she chose to reach out to her friends first because she was more comfortable with them so "that is who I usually talk to about this type of stuff." Maya also said she went to her friends first because, "I just feel more comfortable with my close friends." A few participants said that even though they would seek support from friends, they were concerned about burdening them with their problems when they had their own issues to deal with. Victoria said that she used to talk to her friends but stopped because she felt she was "putting pressure on them because they knew how I was feeling and what I had going on in my life" and she was worried they would "feel bad about me." Ava was also cautious about going to her friends. She was concerned that they were not equipped with the skills or experience to help her. For example, she said "I think most teens will go to their friends, but because we are all teens, I don't think we are giving each other good advice or advice that would actually be helpful." Alejandra shared

that she was "most likely to go to a friend first" but if it was a big issue, she would go to an adult "most likely my mentor or a family member" who might be better prepared to help.

*Trusted adult*. Six participants mentioned that they felt comfortable going to a trusted adult for support. Several of the students said that they had a teacher as an available resource as part of a structured mentor program. Students were matched with their mentor at the beginning of freshman year and remained with their mentor throughout their enrollment at the school. Ava acknowledged that it was very helpful to have a trusted adult at school who she was comfortable talking to. She said that the mentors "are the people we go to if anything is going on academically or even just in our lives in general." She said that she liked her mentor because she felt that her feelings were "validated" which was special to her because "you don't get that from a lot of people." She also said it was helpful to have "someone at school that supports you, and you trust." Alejandra said she was very comfortable going to her mentor and felt that students knew that their mentors or "anyone else in the staff that you really like and trust" were available to them. Maya thought her mentor was "super chill" and liked that she frequently reached out to her students with "checkups where she asked how we are doing with our work." She felt that her mentor supported her by helping to "get some pressure off her" which reduced her stress.

Gabriela mentioned that the tutors in her school-based tutoring program were additional trusted adults she could go to for help. She said that the tutors provided "another support system." She liked them because they were "interactive, open minded, very calm, very chill" and they made her "feel comfortable."

*School counselors*. Although a few students mentioned that they had school counselors at their school, they did not feel that they were accessible for mental health support. Victoria said she knew there was a school counselor but had only spoken to her once. She admitted that she felt uncomfortable talking to the counselor because "you have to go up to them. They don't go up to you, you know, and sometimes kids are just very shy or just don't want to speak out." Maya said her school had a counselor but she had "never been to them." Amelia said she knew there was a counselor "that you could talk to anytime as long as she is not talking to someone else already" but had not contacted her for mental health support.

*Family*. Five participants said they would reach out to a family member for support. Lucia said she was close to her cousin and liked to go to her for support because she felt she could "tell her everything." Victoria liked talking to her brother and found that it was easier to talk to him because "he is older than me and he has already been through it and I just trust him a lot" and because she knows that "he does care for me." Ava, Lucia, Olivia and Alejandra said they would sometimes go to their parents, often mom. Alejandra said she liked to go to her mom because she would try to "comfort me and be supportive."

*Self*. Several participants shared that they preferred to try and manage their feelings on their own rather than seek support from someone else. For example, Alejandra said that there were certain situations where she wanted to reach out to an adult but most of the time she would prefer to "just hold it in, like kind of deal with it." Amelia said she would "keep it to myself" and would prefer to "just figure it out on my own." Victoria also said her preference was to "keep it to myself" rather than share her feelings with someone else.

*Therapy*. When the students were asked about the types of mental health services or resources they used outside of school the only mental health resource they mentioned was therapy. Two participants said they were currently in therapy and three participants had been to a therapist in the past but were no longer seeing a therapist. The participants provided mixed responses about their experiences with therapy. Lucia admitted that even though the "therapist was forced on me" and it "wasn't a choice", in the end, it worked out for her. She said, "I have actually learned a lot of new things" and she went on to say that her experience

was positive and "I actually like her". Ava did not have a good experience in therapy. She said she did not like it because "the trust isn't there and sometimes the things they tell you to do does not really help". Only one student mentioned that their school had a licensed therapist, however, they were not available to everyone. Lucia explained that her school therapist could only see some people and that not everyone "can make an appointment to see her" so she did not feel like she was an accessible resource.

**Program ideas.**   The last interview question was about which features should be included in a new program to motivate students to get help with their anxiety. The students had a lot of ideas which were categorized into three themes: education & coping skills, access to a trusted adult, and interactive games & activities. In addition, the participants had additional considerations that were important to them but did not clearly fit into one of the three themes, which were categorized under a theme called *Things to Consider*. Importantly, all eight students assumed that a program would include a technology component. For example, Ava said that since "most teens use technology" it would "really help the program" to incorporate technology. Amelia suggested creating a "website you could trust."

*Education and coping skills.* All but one student specifically mentioned that a new program should include educational information about anxiety and mental health issues with suggestions on different types of coping skills that could be used for certain situations. Ava felt that everyone needed to be more educated on mental health issues and "how many disorders there are." Amelia wanted a list of coping strategies that she could access based on what was causing her anxiety, like friendships. She suggested these instructions, "once you click one thing on the list, under that there will be information about that coping mechanism [and] how to use it." Olivia also thought that coping mechanisms should be included because she thought teens needed to learn "good coping mechanisms" so they would avoid seeking out unhealthy coping strategies. She said that learning effective coping strategies was important because "a lot of kids like feel like they are alone and they can't trust anyone and so they feel they need to use some bad coping mechanisms." She also thought teens should learn these coping strategies now because it would "help a lot in the long run because they can use that when they are adults."

*Access to a trusted adult.* Six participants thought that a new program should include access to a trusted adult, however, there were different ideas on how that trusted adult should be involved. For example, Victoria said that a lot of teens were afraid to speak with someone so she suggested creating an "app where you can just write how you feel without anybody having to judge you" but she also felt strongly that "there should always be someone to talk to" if a teen wanted to talk to someone. Olivia also thought that a new program should include access to someone to talk to, but she believed teens would prefer that it was an anonymous interaction due to trust issues and comfort level because sharing "depends on how well you know the person." Alejandra agreed that if it was anonymous then she would be "100% comfortable with that."

*Interactive games & activities.* Several participants suggested that a new program should include interactive games and activities such as online word puzzles and coloring. Lucia said she would like a game but suggested adding to it, "I like playing games but then questions at the same time. Questions about yourself and how you are feeling and stuff like that." Maya suggested a feature that would provide "ideas for stuff you could do in a day to just keep you busy." Specifically, she said she would like to learn "new stuff to take my mind off things." Victoria also liked the idea of learning new skills. She said if she were to design something to help kids, "it would probably be a game" and it would be "useful" and "not just teach you, but something that can guide you" and it cannot feel like schoolwork.

*Things to consider.* This theme was created because a few participants wanted to provide some extra advice about how to engage students with a new program. For example, Gabriela

said to stay away from certain words that were unappealing to teens. She did not like the words "anxiety or depression, I feel like it scares me away." She would be interested in a website or app if it used "softer words" and if it had a "place where you could talk to people or a place we could have fun or play games." Gabriela also shared that teens want honest and direct information, no sugarcoating,

> [S]ugarcoating things with younger people nowadays isn't going to be working as effectively as it might have worked in the past because people are more aware of their surroundings due to a lot of social media being introduced at such a young age.

Ava also mentioned words she did not like. She suggested avoiding the words meditation, mindfulness, and therapy because they are a "turn off" for teens. She also did not like the term "mental health issues" but felt "there is not another way to say it. I try to avoid it but it's hard." Maya's advice was to incorporate a social media influencer (an individual with a large social media following) that was relatable to teens. She shared that she followed a young woman from a "Hispanic household." She explained what she liked about this influencer in more detail:

> I feel like she is a very good person to look up to, in my opinion. I feel like I relate to her a lot and I feel like that is what's mostly helped me because she was just talking about how she grew up and how she deals with things and she shows both good and bad sides, like keeps it real with all her supporters. I think that is really inspiring to me because there's always hope to look forward, there is always like a better outcome even if things are going bad.

### Design Thinking workshop

Eleven Hispanic students between the ages of 14 and 17 participated in a 90-minute Design Thinking workshop held after school in the conference room of a small high school with a large Hispanic population. Eight of the participants were female and three were male. One student from the key informant interviews also participated in the workshop. Snacks were provided and each student received a $25 gift card for participation.

*Workshop activities and results*. Students participated in several structured activities commonly used in the Design Thinking process (Table 3). First, they were asked to review the problem statement and persona. Then they were asked to brainstorm all the tools or strategies they thought the persona, Sofia, could use to manage their anxiety. They used post-it notes to record their ideas and posted them on the wall (Fig 3). A total of 98 responses were grouped into 19 categories. The most common coping strategy suggestions, defined as receiving more than five responses, were physical activity (n = 16), self-care (n = 14), talking to someone you trust (n = 11), listening to music (n = 9), doing art (n = 7), shifting mindset (n = 7), breathing/meditation (n = 5), and games/hobbies (n = 5).

Next, students were asked to list the barriers that they thought would prevent the persona from getting help. Students were asked to write as many responses as they could think of in three minutes. Students posted 51 responses. After all responses were posted, each student was given three dot stickers to vote on the three barriers that they felt were the most significant. Based on their votes, the top three barriers that prevented the persona from seeking help were lack of trust (n = 14), fear of judgment (n = 13), and stigma (n = 8).

After identifying the three most significant barriers, students were asked to start thinking about possible solutions. To guide the next activity participants created a "How Might We" statement. "How Might We" (HMW) statements are a Design Thinking tool used to reframe

**Table 3. Design Thinking workshop activities.**

| Activity | Goal | Description |
|---|---|---|
| **Icebreaker: 30 Circles** | Introduce participants to creative thinking | Participants receive a sheet of paper with 30 circles. They are asked to draw as many objects as possible using the circles in 3 minutes. *No further instructions were provided.* |
| **Ideate: Brainstorm** | Exploration of key themes from the key informant interviews, present the problem statement and persona, and prioritize what is most important | Participants responded to the following prompts and voted based on what they felt was most important: <br> 1. What tools work for helping teens with anxiety? <br> 2. What does not work? What are the barriers? What tools do you NOT like? <br> 3. VOTE on what you think are the biggest and most challenging barriers? <br> 4. Identify the number one challenge and create a *How Might We* statement <br> 5. Write as many solutions as possible to solve for these challenges Step 6 (6 mins) <br> 6. VOTE on the best solutions for the challenges |
| **Mind Mapping: visual representation of themes** | Synthesis of key themes from the key informant interviews and individual experience as applied to the persona | Participants were instructed to write the word *Anxiety* in the middle of a blank piece of paper with lines to: <br> • Technology <br> • People who support me <br> • School <br> • Coping strategies <br> then brainstorm ideas from these themes considering the persona as the person seeking support |

the challenges into opportunities and help workshop participants shift to activities that are more focused on problem solving [25]. The workshop participants came up with the following statement:

> ***How Might We*** *design a program or product for Sofia that inspires trust and reduces stigma, and that is inclusive of everyone.*

The students were asked to come up with as many ideas as possible that would help solve the HMW statement and address the top three barriers previously identified. There were 33 responses which were grouped into two categories, *connection with others* and *coping strategies*. The responses in the *connection with others* category included ideas such as "having a mentor or trusted adult to talk to" and "having someone you know you can trust to hang out for a bit." Ideas from the *coping strategies* category included actionable strategies such as

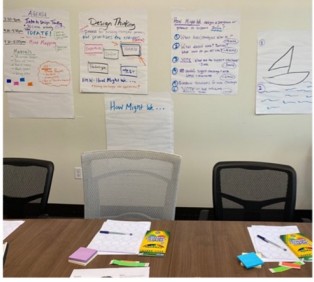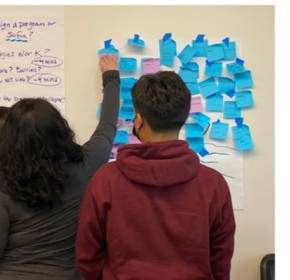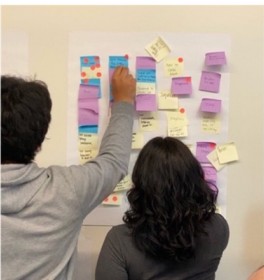

**Fig 3. Student brainstorm session (Ideate) from the Design Thinking workshop.**

download "an app with breathing exercises" and find a "calendar and place to write how you felt on that day." Each student was allowed six votes to assign to their favorite solutions. The ideas with the most votes were: 1) teach different ways to calm down (n = 7), 2) allow people to text someone for support (n = 5), 3) find quiet space that is not home (n = 4), 4) access to a mentor or trusted adult (n = 4) and 5) spending time with someone you trust (n = 3).

**Mind mapping.** As the final workshop activity in the design thinking workshop, each student was asked to draw a mind map using the persona as a reference. The individual mind maps were consolidated and used to create one aggregated meta- map (Fig 4). The large circles that extend from the 4 main themes indicate that there were many students with the same or similar responses. Smaller circles represent fewer responses for that theme. The diamond shapes extending from the School and Technology themes indicate a negative association with that theme.

## Study limitations

There were several limitations to this study. First, the findings may have limited generalizability. The goal was to recruit participants from a variety of different types of schools and sizes (private, public etc.), however, due to a variety of factors that impacted recruitment, most of the students came from the same school. Although their access to school resources were similar and not necessarily representative of other educational settings, their personal experiences with anxiety were unique and they provided diverse views about their help-seeking behaviors. Second, the sample size was small, which was influenced by recruitment barriers related to school policies and lack of relationship with the researcher. Despite the small sample size, there was saturation with codes and themes from the students enrolled at the same school, however, new themes would likely emerge with participants enrolled at larger public schools.

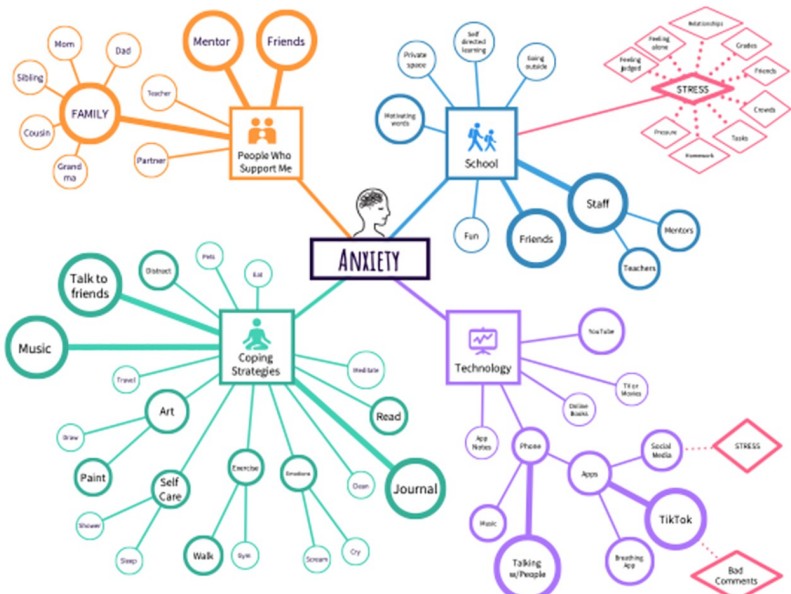

**Fig 4. Consolidated student mind map from the Design Thinking workshop. Description:** Meta map of all responses. The themes are in Squares. The circle sizes are related to the number of students with the same response. The larger circles and thicker lines represent more student responses. The red diamonds represent a negative association with that theme.

Furthermore, all interview participants were young women, highlighting the need to engage more young men, which is a well-known research challenge. Finally, self-selection bias may have occurred because all study participants were volunteers.

## Discussion

The purpose of this research was to gain a deeper understanding of the factors that influence adolescent help-seeking behaviors for the treatment of anxiety. The focus was to work with Hispanic youth to understand the barriers to treatment and to identify program components that would encourage more adolescents to seek help. The results show that the primary reasons adolescents were not seeking help were lack of trust, fear of being judged by others including family and friends, and a negative perception of therapy and therapists.

Although there are several studies that discuss the barriers facing low-income Hispanic youth who need mental health services, to our knowledge there has been no study that has used a Design Thinking research process to engage adolescents in a participatory, solutions-oriented approach to solve a complex public health issue that directly impacts them. It was easier to recruit students for the workshop activity than the interviews. This is likely because students were not asked to talk about themselves or their personal feelings. The Design Thinking workshop format seemed particularly effective because even though we were addressing a sensitive topic that impacted each of them personally, the concept of designing a program to help someone else (the fictional persona, Sofia) appeared to make students more comfortable which resulted in a considerable number of thoughtful responses. The students were genuinely interested in the topic and remained actively engaged throughout the entire workshop. This research reinforces the need to develop evidence-based programs in collaboration with youth that focus on increasing engagement through cultural inclusion and innovation instead of designing programs for youth without their input.

The study participants expressed a general fear and lack of trust in mental health care and services as significant barriers to seeking services which appeared to be deeply rooted in cultural beliefs and parental perceptions. The research published by Ramirez et al. [11] supports this finding that the mistrust of the mental health care system felt by many Hispanic families is not uncommon and is one of the factors that prevent youth from seeking treatment. A study published by Lu et al. [14] reinforces our findings that a parent's negative perceptions and beliefs about mental health seem to have a strong impact on an adolescent's decision to seek help, providing another barrier to accessing mental health services and resources. This highlights the need for additional parent education and support to equip parents with the knowledge and tools to support their child.

Another significant barrier mentioned by participants was their skepticism towards therapy and therapists. While a few participants reported positive experiences, the majority expressed distrust in therapists and found therapy unhelpful. This aligns with findings in a study by Stafford et al. [20] where adolescents cited lack of trust as a key reason why they were unable to establish a meaningful relationship with their therapist, often perceiving therapy as a waste of time. Despite being skeptical about therapists, participants in our study strongly expressed desire for support from a trusted adult. Similarly, the literature suggests that while adolescents had mixed views about therapists, [32] many preferred to seek help from someone other than a therapist and not have a therapist directly involved in treatment [33]. This raises an interesting question of whether therapist involvement is necessary for all adolescents. Our findings suggest that a trusted adult such as a school counselor, health coach, mentor, or school nurse, who is trained to offer personalized, non-judgmental guidance, could be an effective alternative to traditional therapy for adolescents struggling with anxiety. This approach offers

emotional support and practical tools while addressing the mistrust of traditional therapy. Further research is needed to explore how these mentor-like roles could be designed and implemented to bridge the gap in adolescent mental health support and provide a more relatable and appealing option for those reluctant to seek help.

Technology-based interventions have the potential to improve engagement with mental health treatment by increasing accessibility and by reaching more people who need support. Some studies suggest that technology-based interventions can be as effective as face-to-face therapy in addressing youth anxiety issues [21–24]. However, despite these benefits, research shows that technology-based interventions often suffer from low program engagement and completion rates [22, 24]. This is relevant and concerning because engagement in treatment is a critical factor for therapeutic outcomes. Engagement variables such as active participation and program use are strongly associated with improved results [21, 34]. While our study did not directly explore how technology can be incorporated into anxiety treatment, the importance of technology proved to be important across several themes. For example, participants reported already using various forms of technology such as music streaming services, online games and puzzles, videos, and social media as coping strategies and distractions from their anxiety. Importantly, while some distraction and avoidance of anxiety can be effective for coping with stressors, excessive avoidance can become maladaptive and lead to the continued experience of anxiety disorders [35]. Thus, it was encouraging that participants also suggested that technology would be a valuable tool to help them learn more effective skills to manage anxiety beyond using it for distraction. This suggests that technology plays a significant role in how adolescents manage their mental health even if they are not participating in structured treatment. Our findings indicate that while technology holds potential for enhancing mental health support, existing technological solutions such as online therapy do not improve engagement or retention with services. This reinforces the need to develop more innovative youth-friendly, technology-based interventions that go beyond online traditional therapy and incorporate features more aligned with adolescent preferences and coping strategies.

A key objective of this research was to collect concrete ideas directly from adolescents to determine which features should be included in a new program that would motivate more adolescents to seek help for their anxiety. The findings of this study suggest that a new program should be able to deliver culturally inclusive evidence-based content in a way that resonates with youth, provides accessible information and education in a way that is fun and interactive, and includes the involvement of a trusted adult who is not necessarily a therapist. As noted, the exact role of a trusted adult is unclear and warrants more exploration. Studies have shown that the involvement of a trusted adult in a technology-based intervention can provide supportive accountability and motivation leading to increased engagement and longer retention resulting in better health outcomes [36, 37]. When developing a new intervention, special attention should be given to how to incorporate a trusted adult and the role of social media which was frequently mentioned as both a coping strategy and a source of anxiety.

## Conclusions

Despite the significant funding that is being allocated for youth mental health support, the current solutions (e.g., hiring more therapists, offering online therapy, implementing more social-emotional learning (SEL) curricula, or starting more support groups) are not increasing the number of youth and adolescents receiving and completing mental health interventions. To increase engagement and retention with treatment, Hispanic youth need more culturally inclusive programs that reduce structural and psychosocial barriers such as those we identified. Since technology is such an integral part of adolescent lives, there is an opportunity for

additional research on innovative technology-based interventions that can reach priority populations, safely deliver evidence-based education and coping strategies, and improve communication and connections with trusted adults to increase youth engagement with mental health treatment.

## Supporting information

**S1 Table. Codebook from teen interviews.** From key informant interviews.
(DOCX)

## Acknowledgments

We gratefully acknowledge the remarkable youth participants who bravely shared their personal experiences as well as the supportive school administrators who helped facilitate participation and provided a safe space to conduct the Design Thinking workshop.

## Author Contributions

**Conceptualization:** Thea Runyan, Kristen Hassmiller Lich, Karl Umble, Karen Li, Kiersten Daniel Baca, Leah Frerichs.

**Data curation:** Thea Runyan, Karen Li, Kiersten Daniel Baca.

**Formal analysis:** Thea Runyan.

**Investigation:** Thea Runyan.

**Methodology:** Thea Runyan, Kristen Hassmiller Lich, Karl Umble, Karen Li, Kiersten Daniel Baca, Leah Frerichs.

**Project administration:** Thea Runyan.

**Resources:** Thea Runyan, Karen Li, Kiersten Daniel Baca.

**Software:** Thea Runyan.

**Supervision:** Kristen Hassmiller Lich, Karl Umble, Karen Li, Kiersten Daniel Baca, Leah Frerichs.

**Validation:** Thea Runyan.

**Visualization:** Thea Runyan.

**Writing – original draft:** Thea Runyan.

**Writing – review & editing:** Thea Runyan, Kristen Hassmiller Lich, Karl Umble, Karen Li, Kiersten Daniel Baca, Leah Frerichs.

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
