## [Decision Letter · Decision Letter 0]

28 Oct 2024

PONE-D-24-23088“Using a youth-centered approach to design an intervention for the treatment of anxiety in low-income Hispanic youth”PLOS ONE

Dear Dr. Frerichs,

Thank you for submitting your manuscript to PLOS ONE. After careful consideration, we feel that it has merit but does not fully meet PLOS ONE’s publication criteria as it currently stands. Therefore, we invite you to submit a revised version of the manuscript that addresses the points raised during the review process.

We look forward to receiving your revised manuscript.

Kind regards,

Linda Kirsten Tip, PhD

Academic Editor

PLOS ONE

Journal Requirements:

2. We note that Figure 2 includes an image of a participant in the study. 

As per the PLOS ONE policy (http://journals.plos.org/plosone/s/submission-guidelines#loc-human-subjects-research) on papers that include identifying, or potentially identifying, information, the individual(s) or parent(s)/guardian(s) must be informed of the terms of the PLOS open-access (CC-BY) license and provide specific permission for publication of these details under the terms of this license. Please download the Consent Form for Publication in a PLOS Journal (http://journals.plos.org/plosone/s/file?id=8ce6/plos-consent-form-english.pdf). The signed consent form should not be submitted with the manuscript, but should be securely filed in the individual's case notes. 

Please amend the methods section and ethics statement of the manuscript to explicitly state that the patient/participant has provided consent for publication: “The individual in this manuscript has given written informed consent (as outlined in PLOS consent form) to publish these case details”. 

**Additional Editor Comments:**

As Reviewer #1 points out, this research has much to offer. However, they have also highlighted a number of inconsistencies in the manuscripts, and in some places a lack of clarity and lack of inclusion of relevant literature. In order to accept the manuscript for publication, it is required to address all issues mentioned by Reviewer #1. This will ensure that conclusions are presented in appropriate fashion and are supported by the data. Reviewer #2 raises concerns about what the manuscript will contribute to existing academic knowledge in the field. We would like to provide you the opportunity to consult and the literature provided by Reviewer #2, and clarify the contribution of your work in relation to this existing work on a very similar topic. I appreciate that this might be a rather steep hill to climb, but I hope that you will appreciate the opportunity to improve the manuscript to ensure it meets our criteria for publication.

Reviewers' comments:

Reviewer's Responses to Questions

**Comments to the Author**

1. Is the manuscript technically sound, and do the data support the conclusions?

Reviewer #1: Partly

Reviewer #2: No

2. Has the statistical analysis been performed appropriately and rigorously? 

Reviewer #1: N/A

Reviewer #2: No

3. Have the authors made all data underlying the findings in their manuscript fully available?

Reviewer #1: Yes

Reviewer #2: Yes

4. Is the manuscript presented in an intelligible fashion and written in standard English?

Reviewer #1: Yes

Reviewer #2: Yes

5. Review Comments to the Author

Reviewer #1: 1. Main Claims:

This work identifies that Hispanic youth with anxiety challenges want culturally relevant technology-based programs providing easily accessible educational information and coping strategies that are engaging and facilitate the support of a trusted adult.

It is worth noting that the goal of the research is stated as follows: “to better understand the barriers and facilitators to youth engagement with mental health services and to identify the important criteria to consider for the design of and engaging intervention that would encourage Hispanic youth to seek help for anxiety and keep them engaged with treatment.” The assumption is that they mean Hispanic youth in the US, since this would be different if it were a study conducted in a Central or South American society, for instance. For clarity, that caveat should be included in the statement of the goal of the research.

This goal is stated as about engagement in treatment, not treatment itself. Yet the title of the paper states that it is about designing a treatment. Please align the title with the actual outcome of the research.

2. Context with Previous Literature:

The discussion of the literature focuses on barriers to accessing treatment for Hispanic youth and low-income youth with anxiety in the US and the implications of no treatment for anxiety.

There is only one reference to the Design Thinking methodology in the Introduction (plus a second one in the methods section). The Introduction would benefit from more elaboration on this approach and the potential strengths and weaknesses of it. This is particularly important since the novelty of the study is described (in the Discussion) as having used this methodology.

3. Data Analysis Support:

This study utilizes Design Thinking as it’s methodological foundation. That involves 5 stages – empathizing, defining, ideating, prototyping and testing. The authors note that they only conducted the first 3 of these: empathizing, defining and ideating. It would be clearer if the authors acknowledge up front that they are not using a full Design Thinking approach with the goal of creating solutions. Rather, they are using the first 3 phases of Design Thinking to better define the problem and identify domains of solutions.

The study is reported to have recruited low income participants. Yet there were no data presented to define what that means or verify that it pertained to each of the participants. I don’t believe that we should assume that all Hispanic youth in a particular school or location are low income, but that does depend on how the construct is defined and that assumption would need to be justified.

The first stage of the project involves key informant interviews about help-seeking behaviors. The section on Key informant interviews has a list of the questions asked, which is helpful. There is mention of a Fig 1 Social Cognition Theory. The first actual figure is Fig 2. This is followed by Figs 3 and 4 and then, finally Fig 1.

The stated goals and methodological details need to influences the title of the paper. It is not the case that the paper describes the “design of an intervention for the treatment of anxiety”. Rather, it describes supports that can be useful to help engage Hispanic youth with anxiety. Treatment is more extensive and can be defined more specifically than the supports identified by the qualitative methods used here. It is noteworthy, for example, that treatment for anxiety involves a great deal of psychoeducation about the purpose and nature of anxiety, and then emphasizes that avoidance of things that cause anxiety lead to a worsening of anxiety. This dilemma – that people want to avoid what makes them anxious but that worsens their anxiety – is at the heart of resolving anxiety through treatment. It is worth considering how much youth, in particular, overlook this and focus only on making the anxiety go away in the moment or taking their attention off of it momentarily.

Table 1 has some inconsistencies that need correction. Stage 2 Purpose does not really seem to be “Challenges & Pain points” though that is the text in that box. Rather, the purpose seems to be “define the problem” which is in the box defining the method of that stage.

The text describing Stages 1 and 2 start with a sentence on the objective of that stage. The text describing Stage 3 should also begin this way. The description of stage 3 makes it clear that “treatments” are not the intent of that research. Identifying appropriate coping strategies and support identification is what was defined by the research process. While this is also important, it is not treatment so should not be conflated as such. The exclusion criteria ensured that depressed, self-harming and suicidal individuals were excluded. This is consistent with a research design to finding coping strategies and supports. With the less severely ill population, coping strategies and supports are appropriate. But the writing of each section, including the title and the abstract, need to reflect this so that the reader is not expecting to hear about treatment interventions.

It would be helpful to know if and to what extent there was overlap in the student-participants across the different activities (stages) of the study.

In the Discussion the mention of technology is raised. There is acknowledgement that the youth in this study liked to use technology for “distraction from their anxiety” but evidence in the literature shows that technology fails to improve engagement and retention in treatment. These are not contradictory, since distraction and treatment are different. Again, the distinction between treatment and coping mechanisms (including distraction) are blurred in the manuscript and need to be clarified. Since the primary treatment for anxiety is reducing avoidance it is possible that it will be difficult for a technological solution to be forthcoming, though research on that topic would be timely. It is also unlikely that a young person would spontaneously recognize that reducing and eliminating avoidance is the solution to their anxiety challenges without having engaged in some actual treatment. It certainly did not come up in the conversations with youth described here.

Overall, then, this research has much to offer. Yet the description of it provided here has some inconsistencies that need to be clarified.

Reviewer #2: The Manuscript "Using a youth-centered approach to design an intervention for the treatment of anxiety in low-income Hispanic youth" (PONE-D-24-23088) does not represent a significant contribution to the literature.

1. Redundancy in Target Population and Culturally Adapted Interventions

The manuscript focuses on developing a culturally relevant, youth-centered intervention for low-income Hispanic adolescents with anxiety. However, similar interventions have already been developed and published. For instance, the "Streamlined Pediatric Anxiety Program" (SPAP) has effectively designed and implemented an anxiety intervention tailored for Hispanic youth in school settings (Pina et al., 2023). SPAP utilized user-centered design and participatory research methods to adapt evidence-based interventions (EBIs) for real-world application among diverse populations, including low-income Hispanic communities.

Additionally, the "Family Intervention for Resilience and Strengthening" (FIRST) program has demonstrated success in culturally tailoring anxiety interventions for Hispanic families (Weersing et al., 2017). FIRST integrates family-based support, addressing cultural nuances and family dynamics pertinent to Hispanic populations. The PLOS ONE manuscript does not offer novel cultural insights or methods beyond what these established programs have already achieved.

2. Established Digital and Technology-Based Intervention Models

The manuscript proposes a technology-based program to engage Hispanic adolescents in managing anxiety. However, this approach is not novel, as existing programs have successfully integrated digital tools into anxiety interventions. The SPAP incorporates gamification and digital resources to enhance engagement and accessibility for youth (Pina et al., 2023). Similarly, digital interventions targeting adolescent anxiety have been explored extensively, with studies demonstrating their effectiveness in increasing engagement and reducing symptoms (Hollis et al., 2017; Wozney et al., 2018). Therefore, the manuscript's digital approach does not constitute a significant advancement over existing literature.

3. Comprehensive Addressing of Psychosocial Barriers

The identification of psychosocial and structural barriers to mental health service utilization among Hispanic youth is well-documented. Both the SPAP and FIRST programs have addressed these barriers by incorporating strategies to reduce stigma, enhance trust, and improve accessibility (Pina et al., 2023; Weersing et al., 2017). The manuscript under review reiterates these known barriers without providing new strategies or insights to overcome them, resulting in a lack of significant contribution to the field.

4. Lack of Theoretical Innovation

The manuscript relies on Social Cognitive Theory and participatory research methods, which have been extensively applied in prior studies. The SPAP, for example, used a "small theory" approach integrating cognitive-behavioral strategies and user-centered design (Pina et al., 2023). The FIRST program also employs evidence-based theoretical frameworks in its intervention design (Weersing et al., 2017). The manuscript does not offer novel theoretical perspectives or methodological advancements beyond these established frameworks.

5. Proven Efficacy of Existing Interventions

Existing interventions like SPAP and FIRST have demonstrated efficacy through rigorous evaluations, including randomized controlled trials. SPAP showed significant improvements in self-efficacy, reductions in anxiety symptoms, and high levels of participant engagement (Pina et al., 2023). FIRST has similarly shown positive outcomes in anxiety reduction among diverse youth populations (Weersing et al., 2017). The PLOS ONE manuscript does not provide evidence that its proposed intervention would yield superior or additional benefits compared to these established programs.

6. Redundancy in Identified Themes and Strategies

The themes identified in the manuscript—such as barriers to seeking help, preferred coping strategies, and the importance of culturally relevant interventions—have been extensively explored in existing literature. The SPAP addresses these themes by integrating them into a streamlined, practical program suitable for school settings (Pina et al., 2023). Reiterating these established findings without adding new knowledge diminishes the manuscript's contribution.

Conclusion

In summary, the PLOS ONE manuscript does not present a significant or original contribution to the field of pediatric anxiety interventions. The proposed approaches and findings largely replicate existing work, particularly those of the SPAP and FIRST programs, which have already effectively addressed anxiety in low-income Hispanic youth through culturally tailored, technology-enhanced interventions. Therefore, the manuscript does not meet the criteria for publication in its current form.

References

Hollis, C., Morriss, R., Martin, J., Amani, S., Cotton, R., Denis, M., & Lewis, S. (2017). Technological innovations in mental healthcare: Harnessing the digital revolution. The British Journal of Psychiatry, 211(5), 263–265. https://doi.org/10.1192/bjp.bp.116.195776

Pina, A. A., Stoll, R. D., Holly, L. E., Wynne, H., Chiapa, A., Parker, J., Caterino, L., Tracy, S. J., Gonzales, N. A., & Valdivieso, A. (2023). Streamlined pediatric anxiety program for school mental health services. Journal of Anxiety Disorders, 93, 102655. https://doi.org/10.1016/j.janxdis.2022.102655

Weersing, V. R., Jeffreys, M., Do, M. C., Schwartz, K. T., & Bolano, C. (2017). Evidence Base Update of Psychosocial Treatments for Child and Adolescent Depression. Journal of Clinical Child & Adolescent Psychology, 46(1), 11–43. https://doi.org/10.1080/15374416.2016.1220310

Wozney, L., McGrath, P. J., Newton, A. S., Hartling, L., Curran, J., Huguet, A., & Cappelli, M. (2018). Translating e-Mental Health Research Into Practice: A Systematic Review of e-Health Program Outcomes in Youth. Journal of Medical Internet Research, 20(1), e1332. https://doi.org/10.2196/jmir.9103

6. PLOS authors have the option to publish the peer review history of their article (what does this mean?). If published, this will include your full peer review and any attached files.

Reviewer #1: No

Reviewer #2: No

---

## [Author Response · Author response to Decision Letter 0]

13 Dec 2024

We thank the editor and reviewers for their thoughtful comments and feedback. Please see attached response to reviewer document for detailed responses.

---

## [Editor Report · Decision Letter 1]

22 Dec 2024

Identifying Engagement Strategies for Hispanic Youth with Anxiety: A Youth-Centered, Design-Thinking Approach

PONE-D-24-23088R1

Dear Dr. Frerichs,

We’re pleased to inform you that your manuscript has been judged scientifically suitable for publication and will be formally accepted for publication once it meets all outstanding technical requirements.

In the new year, you’ll receive an e-mail detailing the required amendments. When these have been addressed, you’ll receive a formal acceptance letter and your manuscript will be scheduled for publication.

Kind regards,

Linda Kirsten Tip, PhD, MSc, PhD

Academic Editor

PLOS ONE

---

## [Editor Report · Acceptance letter]

20 Jan 2025

PONE-D-24-23088R1 

PLOS ONE

Dear Dr. Frerichs, 

I'm pleased to inform you that your manuscript has been deemed suitable for publication in PLOS ONE. Congratulations! Your manuscript is now being handed over to our production team.

Kind regards, 

on behalf of

Dr. Linda Kirsten Tip 

Academic Editor

PLOS ONE